# Trustworthy Monte Carlo

**Juha Harviainen**
University of Helsinki
juha.harviainen@helsinki.fi

**Petteri Kaski**
Aalto University
petteri.kaski@aalto.fi

**Mikko Koivisto**
University of Helsinki
mikko.koivisto@helsinki.fi

## Abstract

Monte Carlo integration is a key technique for designing randomized approximation schemes for counting problems, with applications, e.g., in machine learning and statistical physics. The technique typically enables massively parallel computation, however, with the risk that some of the delegated computations contain spontaneous or adversarial errors. We present an orchestration of the computations such that the outcome is accompanied with a proof of correctness that can be verified with substantially less computational resources than it takes to run the computations from scratch with state-of-the-art algorithms. Specifically, we adopt an algebraic proof system developed in computational complexity theory, in which the proof is represented by a polynomial; evaluating the polynomial at a random point amounts to a verification of the proof with probabilistic guarantees. We give examples of known Monte Carlo estimators that admit verifiable extensions with moderate computational overhead: for the permanent of zero–one matrices, for the model count of disjunctive normal form formulas, and for the gradient of logistic regression models. We also discuss the prospects and challenges of engineering efficient verifiable approximation schemes more generally.

## 1   Introduction

Can we trust a result computed by a machine? Putting aside questions about the sensibility of the computational task itself, trust stems from one's belief in the correctness of the selected algorithm, its implementation in software, and execution on hardware. All these ingredients are prone to errors, especially when the task is hard, the algorithm is advanced, and the computations are extensive (distributed and long). Trustworthy computation is one pillar of trustworthy AI [28], and it is an issue particularly when one "outsources" computations to other computational agents, e.g., to millions of Internet users to make use of their idle cycles—some users could be dishonest and return seemingly valid results without running any actual computation [6, 26].

Ideally, the outcome of the computations would be accompanied with a certificate of correctness, or *proof*, the user can *verify*. Furthermore, *verification should be computationally easy* in that it should consume substantially less computational resources than it takes to run the computations from scratch with state-of-the-art algorithms. For many decision and search problems such proofs are available; e.g., a certificate of satisfiability of a given Boolean formula is simply a satisfying truth assignment. For optimization problems, the cost of the found (locally optimal) solution is typically easy to verify, and certificates of optimality are sometimes obtained by linear programming duality. *Certifying algorithms* extend the output accordingly with an easy-to-verify proof [20]. For counting problems, which are central in applications involving probabilistic reasoning, such certificates are typically not readily available, and it is not immediate whether such certificates exist in the first place.

36th Conference on Neural Information Processing Systems (NeurIPS 2022).

In this paper, we initiate research on *verifiable* randomized approximation schemes for weighted counting. Specifically, we investigate Monte Carlo estimators, which approximate a target quantity $\mu$ by a sample-average of random variables $W_1, W_2, \ldots, W_N$ with mean $\mu$ so that

$$\mu \approx \frac{1}{N} \sum_{t=1}^{N} W_t. \tag{1}$$

For an accurate estimate, one may have to draw a large number of samples. Fortunately, the simple way of aggregating the $N$ samples enables easy and efficient parallelization of the computations, particularly if the samples are independent. This motivates studying a setting where one agent, the *verifier*, delegates most of the computations to one or more other, potentially computationally more powerful, agents. When aggregating the results of the delegated computations, how can the verifier know if some agents made errors, whether intentional or spontaneous, in their computations?

To address this question, we build on ideas and frameworks developed in the context of proof systems with probabilistic soundness in complexity theory and cryptography [9, 6, 8, 26, 27]. One of the key ideas is to encode the problem instance at hand by a univariate polynomial $p$ of degree $D$ such that knowing the $D + 1$ coefficients of $p$—or equivalently, evaluations of $p$ at $D + 1$ points—reveals the correct solution to the problem instance. The verifier delegates the task of computing $p$ and receives a *claimed* polynomial $\tilde{p}$. Now, the verifier evaluates both the correct $p$ and the claimed $\tilde{p}$ at a *single* random point $r$ drawn from a range significantly larger than $D$: if $p \neq \tilde{p}$, then $p(r) \neq \tilde{p}(r)$ with high probability, since $p - \tilde{p}$ has at most $D$ roots. Supposing the total time needed for solving the problem with a state-of-the-art algorithm is $T \approx DE$, where $E$ is the time requirement per evaluation of $p$, an optimal verifier time is achieved when $D \approx E \approx \sqrt{T}$. That is, verification is computationally easy in the sense that it consumes essentially only a square root of the resources needed to solve the problem with state-of-the-art algorithms. Various problems can be cast in this way, with only a small overhead in relation to the time requirement of the corresponding non-verifiable computation. Examples range from the matrix permanent and model counting for Boolean formulas [27] to graph coloring and counting small subgraphs [4], and to exact probabilistic inference in graphical models [14].

When extending the framework to Monte Carlo estimators of the form (1), we encounter a new challenge: the randomness and independence of the summands $W_t$. Indeed, if the verifier outsources the generation of random numbers, there is no deterministic correct result that could be verified. We circumvent this obstacle by letting the verifier generate all needed random bits, after which the remaining computational task is deterministic. This leads to another challenge: if the verifier has to generate $N$ random numbers, how can the verifier avoid its running time scaling as $N$, the goal for scaling being rather $\sqrt{N}$? Here we make use of the facts that (i) only $O(\log N)$ independent random bits suffice to generate $N$ *pairwise* independent random bits, and (ii) pairwise independence suffices for concentration results that stem from bounding the second moment of the random variables $W_t$.

The rest of this paper is organized as follows. Section 2 connects the task of designing a proof polynomial to that of finding an appropriate low-degree arithmetic circuit [27]. In Section 3, we present our main contribution, a generic method for obtaining what we call a verifiable randomized approximation scheme (vras). Section 4 extends the method to settings where our arithmetic circuit only approximates a random variable $W_t$. As a running example, we use the problem of computing the permanent of a given matrix, for which practical approximating schemes have been sought in the AI community recently [18, 10]. We also apply the methods to known Monte Carlo estimators for model counting of Boolean formulas given in a disjunctive normal form and for estimating gradients of a Bayesian logistic regression model. In the last section, we connect to recent work on learning and reasoning with arithmetic circuits [25] and probabilistic generating circuits [29]. We also discuss directions for future research.

Although this work is theoretical and leaves experimentation for future works, we include a proof-of-concept Mathematica [11] implementation for the example of Section 3.4 in the supplement.

Throughout the paper, the notation $\tilde{O}(\cdot)$ suppresses factors polylogarithmic in the argument, $[\![P]\!]$ evaluates to 1 if $P$ is true, and to 0 otherwise, and (†) marks that a proof is given in the supplement.

## 2 Proving and verifying with polynomials

We begin by introducing the proof system, synthetisizing ideas from related recent works [27, 4, 14].

## 2.1 Prove and verify

Suppose one computational agent, the *verifier*, wishes to delegate most of the computational task at hand to another agent, the *prover*. Here the prover may consist of multiple agents working independently of each other, thus efficiently parallelizing the delegated computations.

**Definition 1.** Let $A$ be an algorithm with access to another algorithm $B$. We call $A$ a *verifier* and $B$ a *prover*, if given any problem instance, the output of $A$ is correct whenever the output of $B$ is correct,[1] and otherwise $A$ outputs "untrusted" with probability at least $2/3$. We refer to the time requirements of $A$ and $B$ as the *verifier time* and the *prover time*, respectively.

The constant $2/3$ is customary but arbitrary. As we will point out soon, the probability can be efficiently amplified to arbitrarily close to $1$.

For constructing a verifier, the idea is to represent the essence of the computational task as evaluation of a univariate polynomial $p(x) = p_0 + p_1 x^1 + \cdots + p_D x^D$ over a finite field $F$ at some $e \geq D + 1$ distinct points $\xi_1, \xi_2, \ldots, \xi_e \in F$, selected so that given the values $p(\xi_1), p(\xi_2), \ldots, p(\xi_e)$—assuming they are correct—the verifier can, with little additional work, solve the original problem. Crucially, the correct polynomial $p$ is specified by the problem instance in a way that is known to both the verifier and the prover. From the verifier's viewpoint the procedure is as follows.

### Algorithm V

**V1** Send the prover the problem instance and the points $\xi_1, \xi_2, \ldots, \xi_e$.

**V2** Receive from the prover a proof, i.e., a claimed value $y_k$ of $p(\xi_k)$ for each $k = 1, 2, \ldots, e$.

**V3** Find the coefficients of $\tilde{p}(x) = \sum_{k=0}^{D} \tilde{p}_k x^k$ such that $\tilde{p}(\xi_k) = y_k$ for all $k = 1, 2, \ldots, e$.

**V4** Draw a random point $\xi_0 \in F$ and evaluate $\tilde{p}(\xi_0)$ and $p(\xi_0)$.

**V5** If $\tilde{p}(\xi_0) = p(\xi_0)$, then accept the proof and consume the values $y_k$; otherwise reject the proof.

Note that if $\tilde{p} \neq p$, then $\tilde{p}(\xi_0) = p(\xi_0)$ with probability $D/|F|$, which we can make arbitrarily small by using a finite field $F$ that is sufficiently large in relation the degree $D$.

Several remarks are in order. While $D + 1$ evaluations would suffice for recovering the polynomial, we allow $e \geq D + 1$ to enable error-correction. Indeed, as long as at least $(e + D + 1)/2$ of the $e$ evaluations are correct, the correct polynomial can be efficiently recovered using a Reed–Solomon decoder (see [4] and references therein); e.g., putting $e = 2(D + 1)$, one fourth of the prover's evaluations can be incorrect. The procedure also enables the prover to run the $e$ evaluations in parallel on $e$ computing units without any communication among them. Furthermore, the protocol is, in essence, non-interactive: the verifier only sends one message to the prover and receives one. Finally, in step V3 the computational efficiency of the procedure stems from fast interpolation of univariate polynomials; this is incorporated in Theorem 1, given in the next section.

## 2.2 Multipoint circuit evaluation

For designing the polynomial and selecting the evaluation points, we adopt Williams's [27] generic method, with some changes. The idea is to represent the problem at hand as a task of evaluating an arithmetic circuit at multiple points. This approach differs from that of Björklund and Kaski [4], who craft their polynomials separately for each problem they consider to obtain a fine-grained optimization of the running time.

An *arithmetic circuit* $C$ in variables $x_1, x_2, \ldots, x_n$ over a field $\mathbb{F}$ is a directed acyclic graph, in which each vertex is called a *gate* and labeled by $+$ (sum) or $\times$ (product) if its in-degree is not zero, and otherwise by one of the variables $x_i$ or by a constant from $\mathbb{F}$. Every gate $v$ *computes* a polynomial $C_v \in \mathbb{F}[x_1, x_2, \ldots, x_n]$ in an obvious way. Unless stated otherwise, the circuit is assumed to have a single *output* gate $v$ (i.e., of out-degree zero) and $C$ is identified with the polynomial $C_v$. The *size*,

---

[1]We understand a problem as a map $\Pi$ from a set of instances to a set of solutions, and that given an instance $I$, an output $S$ of an algorithm for $\Pi$ is correct if and only if $\Pi(I) = S$. By saying that $A$ has access to $B$ we mean that $A$ can run $B$ to solve instances of a fixed subproblem $\Pi'$, in relation to which an output of $B$ is either correct or incorrect.

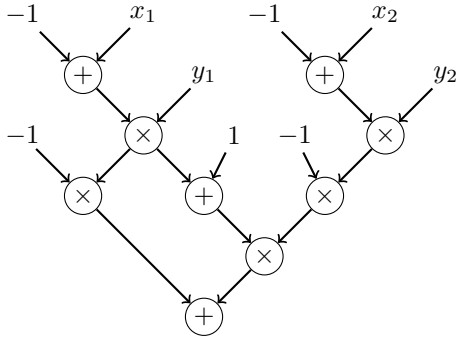

Figure 1: An arithmetic comparison circuit. The circuit outputs 1 if $x < y$, and 0 otherwise. The numbers $x$ and $y$ are given as bit vectors of length $n$; illustrated is the case $n = 2$. The circuit represents the polynomial $\sum_{i=1}^{n}(1 - x_i)y_i \prod_{j=1}^{i-1}\left(1 - (1 - x_j)y_j\right)$ of degree $2n$ using $O(n)$ gates.

*depth*, and *degree* of $C$ are, respectively, the number of gates, the length of the longest directed path, and the largest degree of the monomials in $C$. We assume that the maximum in-degree is at most 2.

Arithmetic circuits are powerful representations of functions and computation. For example, to simulate any Boolean circuit by an arithmetic circuit of about the same size and depth, simply replace each Boolean gate by a constant number of sum and product gates, e.g., $x_1 \lor x_2$ by $x_1 + x_2 - x_1x_2$. Figure 1 shows a circuit for comparing two $n$-bit nonnegative integers.

Typically we want to show that a given function admits a circuit that has small size and low degree. While it is important that we can also efficiently *construct* such a circuit, usually it is clear that construction takes time roughly linear in the size of the circuit, and we do not mention that.

The *multipoint circuit evaluation* problem (MCE) is as follows. Given an arithmetic circuit $C$ in $n$ variables over a *finite* field $\mathbb{F}$ and points $u^1, u^2, \ldots, u^K \in \mathbb{F}^n$, compute the values $C(u^k) \in \mathbb{F}$ for all $k = 1, 2, \ldots, K$. Denote by $s$ and $d$ the size and the degree of the circuit, respectively.

The following result tells that we can design a verifier by representing the problem as an instantiation of MCE. We attribute the result to Williams [27, Theorem 1.2]; we have made some modifications to the proof to also take into account the prover's time requirement (not considered by Williams).

**Theorem 1** (†). *MCE admits a verifier with prover time $\tilde{O}(Kds)$ and verifier time $\tilde{O}(Kdn + s)$.*

*Proof sketch.* The main idea is to transform the circuit $C$ of $n$ variables into a univariate circuit $p$, which then represents a univariate polynomial. We associate each point $u^i \in \mathbb{F}^n$ with an element $\xi_i$ from some large enough extension field $F$ of $\mathbb{F}$. Next, we find an interpolating polynomial for each variable of $C$ such that for a point $\xi_i$ the polynomial $\lambda_j$ extracts the $j$th component $u_j^i$ of $u^i$. As a result, we obtain a circuit with $p(\xi_i) = C(u_1^i, u_2^i, \ldots, u_n^i)$.

The interpolating polynomials increase the degree of the circuit (polynomial) by a factor of $K$, and therefore we need $Kd + 1$ evaluations to determine the polynomial representation of $p$ uniquely. As a consequence, we evaluate $p$ on some additional arbitrary distinct points $\xi_{K+1}, \xi_{K+2}, \ldots, \xi_{Kd+1}$.

The prover needs to evaluate a circuit of size $s$ on $Kd + 1$ points, which takes time $O(Kds)$. On the verifier's side, we need to send the circuit of size $s$ to the prover, as well as $n$ input variables of $C$ for each point $\xi_i$ to enable massive parallelization. This and the other steps of Algorithm V can be computed efficiently in given time limits by using fast interpolation and multipoint evaluation of univariate polynomials. □

**Remark 1.** The result relies on the prover's and the verifier's ability to evaluate the circuit in a given point from $F^n$, where $F$ is an extension field of $\mathbb{F}$. For the time requirement of these evaluations, the circuit size $s$ gives an upper bound, which can be overly conservative—indeed, we may replace the term $s$ by the actual time required for computing the value of the circuit at a given point. For example, the univariate polynomial $1 + x + \cdots + x^d$ in one variable has a circuit of size $2d$ and degree $d$, but it can be evaluated in the form $(1 - x^{d+1})/(1 - x)$ using only $O(\log d)$ field operations.

## 2.3 Example: the permanent

The permanent of a square matrix $A = (a_{i,j}) \in \mathbb{F}^{m \times m}$ is given by

$$\mathrm{per}\,A = \sum_\sigma \prod_{i=1}^m a_{i,\sigma(i)} = \sum_{t \in \{0,1\}^m} \prod_{i=1}^m (-1)^{1-t_i} \sum_{j=1}^m t_i a_{i,j}\,,$$

where $\sigma$ runs through all permutations on $\{1, 2, \ldots, m\}$. The second expression is due to Ryser [24]. Assume $m$ is even and put $n := m/2$ and

$$C(t) := \prod_{i=1}^m (-1)^{t_i - 1} \sum_{j=1}^m t_i a_{i,j} \quad \text{and} \quad C_*(u) := \sum_{v \in \{0,1\}^n} C(u_1, u_2, \ldots, u_n, v_1, v_2, \ldots, v_n)\,.$$

Now $\mathrm{per}\,A = \sum_{u \in \{0,1\}^n} C_*(u)$. Write $(-1)^{1-t_i}$ as $2t_i - 1$ to observe that $C_*$ has an arithmetic circuit of size $O(2^n m^2)$ and degree $2m$. By Theorem 1, the permanent admits a verifier with prover time $\tilde{O}(2^m)$ and verifier time $\tilde{O}(2^{m/2})$.

## 3 Verifiable Monte Carlo integration

We next apply Theorem 1 to the computation of Monte Carlo estimators (1).

### 3.1 Multipoint evaluation and pairwise independence

To reduce the Monte Carlo estimator to $K$ circuit evaluations, we simply write the sum over $N$ terms as a double summation:

$$\frac{1}{N} \sum_{t=1}^N W_t = \frac{1}{N} \sum_{k=1}^K V_k\,, \qquad V_k := \sum_{j=1}^{N/K} W_{(k-1)K+j}\,.$$

Naturally, $K$ has to divide $N$. To make this hold, we will assume that $N$ and $K$ are powers of 2.

Now we view each random variable $V_k$ as an output of a fixed arithmetic circuit evaluated at some point $u^k$. In fact, we take the index $k$ itself as the point $u^k$. The circuit is obtained as a sum of $K/N$ evaluations of a smaller circuit $C$, each of which outputs $W_{(k-1)K+j}$ when given $k$ and $j$ as input. Crucially, the output of $C$ is a deterministic function of $(k, j)$, implying that all randomness must be encoded in $C$. Because of this, the $N$ random variables cannot be independent, for then the encoding would require at least $N$ bits, rendering the circuit $C$ large and undermining the pursuit of an efficient verifier. To circumvent this obstacle, we resort to pairwise independence.

To develop these ideas more formally, we begin with the fact that $\ell$ independent random bits suffice for generating $2^\ell - 1$ pairwise independent bits, by taking the exclusive-or of every nonempty subset [22, Lemma 13.1]. We may index by a bitvector $t \in \{0,1\}^\ell$, interpreted as the integer $t_1 + 2t_2 + \ldots 2^{\ell-1} t_\ell$.

**Definition 2.** The *subset-sum expansion* of $x \in \{0,1\}^\ell$ is the bit vector $y \in \{0,1\}^{2^\ell - 1}$ given by

$$y_t := \bigoplus_{i=1}^\ell t_i x_i\,, \quad (0,0,\ldots,0) \neq t \in \{0,1\}^\ell\,.$$

**Fact 2.** *Let $x$ be a uniform draw from $\{0,1\}^\ell$ and $y$ its subset-sum expansion. Then the bits $y_t$ are pairwise independent uniform draws from $\{0,1\}$.*

Clearly, if the index $t$ is given as a bit vector, the bit $y_t$ can be computed by a succinct circuit:

**Proposition 3.** *Given a bit vector $x \in \{0,1\}^\ell$, there is a circuit $C : \mathbb{F}^\ell \to \mathbb{F}$ of size $O(\ell)$ and degree $\ell$ such that $C(t) = y_t$ for all $t \in \{0,1\}^\ell \setminus \{(0,0,\ldots,0)\}$, where $y$ is the subset-sum expansion of $x$.*

*Proof.* The first layer computes $z_i := t_i x_i$ for $i = 1, 2, \ldots, \ell$. The second layer computes the exclusive-or of $z_1, z_2, \ldots, z_\ell$ using the arithmetization $a \oplus b = a(1-b) + (1-a)b = a + b - 2ab$. $\square$

A single random bit $y_t$ being generally insufficient for generating the random variable $W_t$ of interest, we take $n$ independent copies, where $n$ will be set depending on the specific random variable.

**Lemma 4** (†). *Let $C : \mathbb{F}^n \to \mathbb{F}$ be a circuit of size $s$ and degree $d$. Let $N \leq 2^\ell - 1$ be a power of 2. Then, given bit vectors $x^1, x^2, \ldots, x^n \in \{0,1\}^\ell$ and a factorization $N = 2^a 2^b$, there is a circuit $C_* : \mathbb{F}^a \to \mathbb{F}$ of size $O(2^b(\ell n + s))$ and degree $\ell d$, such that*

$$\sum_{u \in \{0,1\}^a} C_*(u) = \sum_{t=1}^{N} C(y_t^1, y_t^2, \ldots, y_t^n),$$

*where $y^i \in \{0,1\}^{2^\ell - 1}$ is the subset-sum expansion of $x^i$.*

## 3.2 Verifiable approximation schemes

So far we have required that the verifier's output is correct if the prover's output was correct (Definition 1). While this applies to computing Monte Carlo estimators (with a fixed sequence of random bits), as shown above, we find it convenient to extend the definition so that the verifier's output is only required to have probabilistic error guarantees. We focus on controlling the relative error; call a random variable $Z$ an $(\epsilon, \delta)$-*approximation* of a real $r$ if $\Pr(|Z/r - 1| > \epsilon) < \delta$.

**Definition 3.** Let $A$ be an algorithm with access to another algorithm $B$. We call $A$ a *verifiable randomized approximation scheme (vras)* if, given an instance and tolerance parameters $\epsilon, \delta > 0$, it outputs an $(\epsilon, \delta)$-approximation whenever the output of $B$ is correct, and otherwise $A$ outputs "untrusted" with probability at least $2/3$. We refer to $A$ and $B$ as the *verifier* and the *prover* and their time requirements as the *verifier time* and the *prover time*, respectively.

If a Monte Carlo estimator is an $(\epsilon, \delta)$-approximation for sufficiently large $N$, we obtain a vras by applying Lemma 4 and Theorem 5. Typically, for averages of 0–1 random variables in particular, such estimators are derived using Chernoff bounds assuming independent samples, which we do not have here. Fortunately, pairwise independence suffices for concentration results based on the second moment; Chebysev's inequality only needs control of the *critical ratio* $\mathbb{E}[W^2]/\mathbb{E}[W]^2$.

**Theorem 5** (†). *Let $W$ be a random variable with the critical ratio at most a known upper bound $R$. Suppose there is an arithmetic circuit $C : \mathbb{F}^n \to \mathbb{F}$ of size $s$ and degree $d$ such that $W = C(x)$ for $x$ drawn uniformly at random from $\{0,1\}^n$. Then the mean of $W$ admits a vras with a prover time $\tilde{O}(T)$ and a verifier time $\tilde{O}(\sqrt{T} + s + d\log \delta^{-1})$, where $T := R \cdot d \cdot s \cdot \epsilon^{-2} \cdot \ln \delta^{-1}$.*

Note that the total work $T$ is essentially—up to the factor $d$ and the hidden polylogarithmic factors— the same as the requirement of the baseline algorithm that does not enable verification. Usually the term $\sqrt{T}$ dominates the verifier's time requirement. Recall that we implicitly assume that the circuit can also be efficiently constructed for the given problem instance. The challenge is, however, rather to show that the random variable $W$ can be generated by a low-degree circuit.

## 3.3 Example: The Godsil–Gutman estimator for the permanent

Let $A = (a_{ij})$ be an $m \times m$ binary matrix and $B = (b_{ij})$ an $m \times m$ matrix where $b_{ij}$ are independent uniform draws from $\{-a_{ij}, a_{ij}\} \subset \{-1, 0, 1\}$. One can show that $\mathbb{E}\left[(\det B)^2\right] = \text{per} A$. Employing this observation, the Godsil–Gutman estimator [7] for the permanent is given by $N^{-1} \sum_{t=1}^{N} W_t$, where $W_t$ are copies of $W := (\det B)^2$. The critical ratio of $W$ is at most $3^{m/2}$ [15].

We show that the permanent admits a vras. By Theorem 5, it suffices to give an arithmetic circuit $C$ that computes $W$ given some number $n$ of random bits as input. We put $n := m^2$ so that there is one random bit $x_{ij}$ per matrix entry. The first layer of $C$ constructs $B$ by $b_{ij} := (1 - 2x_{ij})a_{ij}$. The second layer computes the determinant of $B$ using, e.g., Berkowitz's division-free circuit of size $O(m^4)$ [3]; in fact, by Remark 1, we may use division and replace the bound by $O(m^3)$. The last layer raises the obtained result to the second power. Since the degree of the determinant is $m$, the degree of the circuit is $2m$. By Theorem 5, the prover time and verifier times are $\tilde{O}(T)$ and $\tilde{O}(\sqrt{T})$, where $T = 3^{m/2}\epsilon^{-2}\ln \delta^{-1}$, ignoring a factor polynomial in $m$. The circuit is defined over the field $\mathbb{Z}/q\mathbb{Z}$, where $q$ is a large prime; it is sufficient to have $q \geq 2N(n!)^2$ so that the arithmetic does not overflow. Large primes can be efficiently found [2].

### 3.4 Example: A quaternion based estimator for the permanent

A more sophisticated Godsil-Gutman type estimator [5] uses a random matrix $B$ whose entry $b_{ij}$ is zero if $a_{ij} = 0$, and otherwise an independent uniform draw from the signed quaternion basis $\{\pm 1, \pm i, \pm j, \pm k\}$. Representing the four basis elements, respectively, by the complex-valued $2 \times 2$ matrices $\begin{pmatrix} 1 & 0 \\ 0 & 1 \end{pmatrix}$, $\begin{pmatrix} i & 0 \\ 0 & -i \end{pmatrix}$, $\begin{pmatrix} 0 & 1 \\ -1 & 0 \end{pmatrix}$, and $\begin{pmatrix} 0 & i \\ i & 0 \end{pmatrix}$, and letting $D = (d_{ij})$ be the corresponding $2m \times 2m$ matrix representation of $B$, the random variable of interest is $W := \det D$. The mean of $W$ equals $\operatorname{per} A$ and the critical ratio of $W$ is at most $(3/2)^{m/2}$ [5], improving on the basic estimator.[2]

Accordingly, we can apply Theorem 5 to show that the permanent admits a vras with a prover time $\tilde{O}(T)$ and a verifier time $\tilde{O}(\sqrt{T} + \log \delta^{-1})$, where $T = (3/2)^{m/2} \epsilon^{-2} \ln \delta^{-1}$. Now our circuit to compute $W$ takes as input 3 random bits $x, y, z$ per entry $a_{ij}$ of $A$ and puts

$$\begin{pmatrix} d_{2i-1,2j-1} & d_{2i-1,2j} \\ d_{2i,2j-1} & d_{2i,2j} \end{pmatrix} := a_{ij}(1 - 2x)\big[y\big(zM_1 + (1-z)M_2\big) + (1-y)\big(zM_3 + (1-z)M_4\big)\big],$$

where $M_1, M_2, M_3, M_4$ are the $2 \times 2$ matrices given above. Thus the degree of each entry of $D$ is 3, whence the degree of $\det D$ is $6m$. The circuit is defined over $\mathbb{Z}/q\mathbb{Z}$, with an appropriate prime $q$: we require $q \geq 2N(n!)^2$ to avoid overflows and, additionally, that $q = 1 \bmod 4$ to guarantee that the field has fourth roots of unity (to serve as $i$, $-1$, $-i$, and 1); the roots can be efficiently found [1].

## 4 Extensions for approximate circuits

So far we have assumed that the random variable $W$ can be *exactly* computed by a succinct arithmetic circuit. We next extend the method to allow for circuits that only *approximate* the random variable.

### 4.1 Small deviation with high probability

Clearly, to obtain an $(\epsilon, \delta)$-approximation, the expected value of the circuit need not be exactly that of $W$— achieving a relative error of order $O(\epsilon)$ should suffice. Likewise, the shape of the distribution need not be similar to that of $W$—having a reasonable upper bound for the critical ratio should suffice. Moreover, we may let the circuit fail and not deliver these features with a small probability or order $O(\delta)$. We next formalize this intuition.

For a random variable $Y$ and event $A$, write $\sigma[Y|A]$ for the standard deviation of the conditional distribution, i.e., for the (nonnegative) square root of the conditional variance $\mathbb{E}[Y^2|A] - \mathbb{E}[Y|A]^2$.

**Definition 4.** Let $(\Omega_i, \mathcal{F}_i, P_i)$ be probability spaces and $X_i : \Omega_i \to \mathbb{R}$ random variables, $i \in \{1, 2\}$. Let $\epsilon, \delta \in [0, 1]$. We say that $X_2$ is an $(\epsilon, \delta)$-*deviate* of $X_1$ if there exists an *acceptance event* $A \in \mathcal{F}_2$ such that $P_2(A) \geq 1 - \delta$ and $\mathbb{E}[X_2|A]$ and $\sigma[X_2|A]$ approximate $\mathbb{E}[X_1]$ and $\sigma[X_1]$ to within a relative error of $\epsilon$ and 1, respectively.

Our choice of the upper bound 1 for the relative error of the standard deviation is somewhat arbitrary. One may replace it by any small number, incurring only a respective small factor to the running time.

**Corollary 6** (†). *Let $W$ be a random variable with the critical ratio at most a known upper bound $R$. Suppose that, for any $\gamma := (\alpha, \beta) \in (0, 1)^2$, there is an arithmetic circuit $C_\gamma : \mathbb{F}^{n_\gamma} \to \mathbb{F}$ of size $s_\gamma$ and degree $d_\gamma$ such that $C_\gamma(x)$ is an $(\alpha, \beta)$-deviate of $W$ when $x$ is drawn uniformly at random from $\{0, 1\}^{n_\gamma}$. Then, for any $\alpha \leq \epsilon/3$ and $\beta \leq \delta\epsilon^2/(3456R\ln(2/\delta))$, the mean of $W$ admits an $(\epsilon, \delta)$-vras with a prover time $\tilde{O}(T)$ and a verifier time $\tilde{O}(\sqrt{T} + s_\gamma + d_\gamma \log \delta^{-1})$, where $T := R \cdot d_\gamma \cdot s_\gamma \cdot \epsilon^{-2} \cdot \ln \delta^{-1}$.*

We have not optimized the constants. Note also that we expect the parameter $\beta$ to affect the circuit size and degree only logaritmically, i.e., by a factor of $(\ln \beta^{-1})^{O(1)}$; see below for an example.

### 4.2 Example: Approximate model counting of disjunctive normal forms

Given a Boolean formula $\psi$ of $n$ variables in a disjunctive normal form $C_1 \vee C_2 \vee \cdots \vee C_m$, where each clause $C_j$ is a conjunction of literals, the DNF counting problem asks the number of truth

---

[2]The random variables $(\det B)^2$ and $\det D$ share the same first and second moments, but the former does not admit efficient computation due to non-commutativity of the quaternion algebra [5].

assignments that satisfy $\psi$, denoted by $c(\psi)$. The problem is a fundamental #P-complete with applications in diverse domains ranging from network reliability to probabilistic databases (see, e.g., [21] and references therein). We next present a verifiable extension of a classic Monte Carlo algorithm [16, 22, Sect. 10.2] that outputs an $(\epsilon, \delta)$-approximation in time $O(m^2 n \epsilon^{-2} \log \delta^{-1})$.

Let $S_j$ be the set of truth assigments that satisfy $C_j$. Generate a random variable $W$ as follows: first select $j$ with probability proportional to the cardinality $|S_j|$; then draw an assignment $a \in S_j$ uniformly at random; let $W := 1$ if $a$ does not belong to $S_k$ for any $k < j$, and $W := 0$ otherwise.

**Lemma 7** ([16], †). *The mean of $W$ equals $c(\psi)/\sum_{j=1}^m |S_j|$ and the critical ratio is at most $m$.*

As computing the sum of the sizes $|S_j|$ is easy, it remains to give a succinct arithmetic circuit that computes $W$. The challenge is to turn given $\ell$ random bits into a sample from the categorical distribution where the probability of $j \in \{1, 2, \ldots, m\}$ is proportional to $|S_j|$. We use rejection sampling. When the random bit string, interpreted as an integer, falls in the range spanned by the sum of $|S_j|$, the sample is accepted and output of the circuit is a copy of $W$. Otherwise, the sampling fails and the circuit outputs 0. The probability of failing can be made small:

**Lemma 8** (†). *For any $\beta > 0$, there is an arithmetic circuit of size $O\left(m(n + \log \beta^{-1})\right)$ and degree $O\left(mn(n + \log \beta^{-1})\right)$ such that the output of the circuit is an $(0, \beta)$-deviate of $W$ when given $O(n + \log \beta^{-1})$ independent uniformly distributed bits as input.*

**Remark 2.** If the mean of $W$ is at most $1/2$ (guaranteed, e.g., by duplicating each clause), then the constructed random variable is also an $(\beta, 0)$-deviate of $W$. Namely, when the sampling of $j$ fails, the output is 0. Thus, the mean of the Bernoulli variable is at most the exact mean $\mu$ and at least $(1 - \beta)\mu$. The variance is at most $\mu(1 - (1 - \beta)\mu) = \mu(1 - \mu) + \beta\mu^2 \leq (1 + \beta)\mu(1 - \mu)$ and at least $(1 - \beta)\mu(1 - \mu)$, implying that also the standard deviation is within a relative error of at most $\beta$.

Having a circuit for $W$, we are ready to apply Corollary 6. For simplicity of expressions, let us make the modest assumption that $\ln \delta^{-1} = \tilde{O}(n)$ *or* $\ln \epsilon^{-1} = \tilde{O}(n)$. By substituting the above size and degree bounds to the formulas in the corollary, we get, after hiding logarithmic factors, that the DNF counting problem admits a vras with a prover time $\tilde{O}(T)$ and a verifier time $\tilde{O}(\sqrt{T} + mn^2 \log \delta^{-1})$, where $T = (mn)^3 \epsilon^{-2} \ln \delta^{-1}$. (We leave the verification of these bounds to the reader.)

This example highlights some new features of our method. Since the required Monte Carlo sample size is relative small (linear in $m$) in comparison to the time requirement per sample (linear in $mn$ in the original algorithm), the verifiable extension did incur a non-negligble overhead in the total complexity, by a factor of $mn^2$. Crucially, however, the cost of the verifier remains significantly lower than the total cost of the original algorithm. In particular, for small $\epsilon$ we have $\epsilon^{-1} \ll \epsilon^{-2}$.

### 4.3 Example: Gradient estimation

In variational inference the posterior distribution of model parameters $w$ is approximated by a parametric distribution $q(w; \theta)$ whose parameters $\theta$ are optimized to maximize the objective

$$F(\theta) := \mathbb{E}_{q(w;\theta)} [f(w)] - \mathrm{KL}(q||p),$$

where $f$ is the log-likelihood of the data, $p$ is a prior of $w$, and KL is the Kullback–Leibler divergence [13]. A popular optimization method iteratively updates the parameters in the direction of the gradient $\nabla_\theta F(\theta)$, for which an approximation is obtained using a Monte Carlo method.

As a concrete instantiation, consider Bayesian logistic regression [12, 23], in which the probability of a response $y \in \{-1, 1\}$ given a feature vector $x \in \mathbb{R}^n$ is modeled as $g(yx \cdot w)$, with $g(z) = 1/(1 + e^{-z})$. The prior $p$ for the parameter vector $w \in \mathbb{R}^n$ is taken as a multivariate normal $\mathcal{N}(0, cI)$ with some constant $c > 0$, and the posterior approximation $q$ is a multivariate normal $\mathcal{N}(\mu, \Sigma)$ with $\Sigma = \mathrm{diag}(\sigma_1, \sigma_2, \ldots, \sigma_n)$. Thus $\theta = (\mu, \Sigma)$ and $f(w) = \sum_{(x,y) \in S} \ln g(yx \cdot w)$, where $S$ is the given dataset of points $(x, y)$, which are modeled as exchangeable (i.e., independent given $w$).

Now, consider the estimation of the gradient of the expectation of $f(w)$; the gradient of the KL divergence can be obtained analytically. The basic estimator is given by

$$\hat{\eta}_N := \frac{1}{N} \sum_{t=1}^N f(w^t) \nabla_\theta \ln q(w^t; \theta), \quad w^1, w^2, \ldots, w^N \sim q(w; \theta).$$

A popular variant, we omit here for simplicity, constructs the estimator based on a subsample of the data set, obtained by random sampling or by deterministically splitting the data set into batches [23].

To apply the method of trustworthy Monte Carlo, we need to find an arithmetic circuit that approximates a random variable $W := f(w)\nabla_\theta \ln q(w; \theta)$, where $w \sim q(w; \theta)$, given some number of uniformly distributed bits. In our example model, one can easily verify that the partial derivatives of $\ln q(w; \theta)$ with respect to $\mu_i$ and $\sigma_i^2$ equal $(\mu_i - w_i)/\sigma_i^2$ and $[(\mu_i - w_i)^2 - \sigma_i^2]/(2\sigma_i^4)$, respectively. These terms being simple polynomials of $w_i$, they can be handled in a straightforward manner, once the scalar $w_i$ has been generated, increasing the size and degree of the circuit very little. The actual challenge is to construct a circuit for $f(w)$. We next sketch a construction in four phases (details in Suppl. B). Since $f(w)$ can be arbitrarily close to $0$ (assuming exact real arithmetic), our goal is to approximate $f(w)$ to within an *absolute* error $\epsilon > 0$ with high probability (over the random $w$).

The circuit takes as input $n$ independent and uniformly distributed bit vectors $r_1, r_2, \ldots, r_n \in \{0, 1\}^\ell$. It operates with numbers of the form $k \cdot 2^{-m}$, where $m$ is a positive integer and $k$ is an integer between $-4^m + 1$ and $4^m - 1$ represented by $2m + 1$ bits; we denote by $\mathbb{B}_m$ the set of these numbers. To enable exact arithmetic, we let $m$ grow as the computations progress in the circuit, yet keeping $m = O(\ell)$. We assume that the data values $x_i, y_i$ and model parameters $\sigma_i, \mu_i$ are represented as elements of $\mathbb{B}_\ell$. We return to the question of how to set $\ell$ after the description of the four phases.

1. *Sampling from the normal distribution.* For each dimension $i = 1, 2, \ldots, n$, a subcircuit maps $r_i$ to a $w_i \in \mathbb{B}_m$ that follows approximately a (discrete) normal distribution $\mathcal{N}(\mu_i, \sigma_i^2)$. We use a finite-precision version of an algorithm that would draw exact samples if we could peform exact arithmetic with real numbers. Each subcircuit is of size and degree $\ell^{O(1)}$.

2. *Computing the dot product.* For each data point $(x, y) \in S$, a subcircuit maps $w \in \mathbb{B}_m^n$ to a $z \in \mathbb{B}_{2m}$ that equals $yx \cdot w$. Each subcircuit is of size and degree $\ell^{O(1)}$, assuming $\ell = \Omega(n)$.

3. *Evaluating the log-sigmoid.* For each data point $(x, y) \in S$, a subcircuit maps the corresponding $z \in \mathbb{B}_{2m}$ to a number $v_{x,y} \in \mathbb{F}$ that approximates $\ln g(z)$ with an absolute error at most $\alpha$; we let $\alpha = O(\epsilon/|S|)$ to keep the total absolute error below $\epsilon$. We use a cubic spline in an interval of length $O(\log \alpha^{-1})$, centered at $0$. Each subcircuit is of size $O(\alpha^{-1/4} \log \alpha^{-1})$ and degree $O(\log \alpha^{-1})$; we assume $m = O(\log \alpha^{-1})$.

4. *Summing over the data points.* The last layer computes the sum $\sum_{(x,y)\in S} v_{x,y}$. This subcircuit is of size $O(|S|)$ and degree $1$.

We argue (Suppl. B) that in phases 1 and 2 the numbers $z$ are generated from a distribution that rapidly approaches the exact distribution as $\ell$ increases. Phase 3 assumes the input numbers $z$ to have precision $O(\alpha)$; we achieve this by setting $\ell, m = O(\log \alpha^{-1})$, which is $O(\log |S| + \log \epsilon^{-1})$.

To bound the size and degree of the circuit, assume for simplicity that $|S| \geq n$. Adding up the size bounds for the subcircuits yields $|S|(\ell^{O(1)} + O(\alpha^{-1/4} \log \alpha^{-1}))$, which is $\tilde{O}(|S|^{5/4}\epsilon^{-1/4})$. Multiplying the degree bounds for the four phases yields $\ell^{O(1)}$, which is $(\log |S| + \log \epsilon^{-1})^{O(1)}$. We conclude, by the proof of Theorem 5, that asymptotically the prover time is $\tilde{O}(T)$ and the verifier time is $\tilde{O}(\sqrt{T} + E)$, where $E := |S|^{5/4}\epsilon^{-1/4}$ and $T := NE$.

This example shows that we can engineer relatively succinct arithmetic circuits to approximate non-polynomial functions. That said, it also shows that representing numbers as bit vectors to enable comparison (in phase 3) incurs a cost that cumulates across the layers in the circuit; e.g., the dot product would require a very low-degree small circuit if the arithmetic was directly over $\mathbb{F}$.

## 5 Concluding remarks

We gave a generic method for equipping Monte Carlo estimators with a certificate of correctness. This method contributes towards general trustworthiness of machine learning and reasoning by enabling safe, verifiable outsourcing of sampling-based integration. When the random variables of interest can be (approximately) represented by low-degree arithmetic circuits, the method incurs only a small computational overhead and the verifiers's work is about a square root of the total work. The framework readily supports massive and simple parallelization of the computations, along with efficient error correction, with negligible additional cost [4, 14]. We used a diverse set of example

estimators to demonstrate the technical versatility of the method, with the understanding that the need for trustworthy computations may vary considerably across real application domains.

The present work being theoretical, many practical issues remain to be addressed through implementation and experimentation. For example, our bounds hide polylogarithmic factors and leave room for "shaving logs" and optimizing constant factors. Another direction is to more systematically engineer building blocks needed for constructing low-degree (approximating) arithmetic circuits; we exhibited a variety of ideas and techniques, but there is a need for a "catalogue" of primitive functions and standard distributions. Finally, there are fundamental questions regarding the limitations of the method: Do random variables that result from sequential importance sampling or rejection sampling (e.g., for estimating the permanent [18, 10]) admit low-degree circuits? Can the method be extended beyond pairwise independence, for example, to Markov chain Monte Carlo?

Given the challenges in engineering low-degree circuits, it would be tempting to *learn* succinct arithmetic circuits from examples, removing the need of human effort. Unfortunately, learning arithmetic circuits appears to be hard already for relatively restricted classes of circuits with black-box access to the function [25]; however, some interesting classes of low-depth circuits do admit efficient learning [17]. Arithmetic circuits have also found their use as tractable probabilistic models. In particular, *probabilistic generating circuits* [29] support efficient inference and subsume related models, such as determinantal point processes (e.g., [19]). Our work suggests that these models might serve as building blocks that readily enable extensions to efficient *verifiable* inference.

## Acknowledgements

This research is partially supported by the Academy of Finland, Grant 316771.

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
