# OpenReview forum: "Trustworthy Monte Carlo"
_NeurIPS.cc/2022/Conference — NeurIPS 2022 Accept_

### Official Review · Reviewer_RDdx · 2022-07-11

**Rating:** 6
**Confidence:** 3
**Soundness:** 3 good
**Presentation:** 3 good
**Contribution:** 3 good

**Summary:**

This paper introduces an MC integration technique in that the computations have an algebraic proof of correctness where the proof is polynomial evaluated at random points. It provides a verified randomized approximation scheme for weighed counting to approximate the target quantity by use of MC.

**Questions:**

- how does it compare to others for instance [27], [8]?
- how is pairwise independence not a very restrictive assumption in practice?
- The performance of an MC estimator depends on the variance of the estimator. How does this method which equips MC estimators with proof correctness depend on the variance of the estimator and how would it be affected?
 -  Even though the gradient estimation for VI is interesting, however, the used estimate of the gradient is shown to have high variance. How does it work for other more advanced gradient estimators and how would the size of sub-circuits change to keep the error low?

**Limitations:**

- Samples must be i.i.d. which is hard to draw in practice.
- Lack of comparison to other techniques
- seems choosing circuits can be difficult and begs some more discussion on that.
- Even though some nice examples are provided it feels like it is not something widely applicable specifically for state-of-the-art methods.

**Strengths And Weaknesses:**

Strengths:
- It is well-written and easy to follow.
- Proof of this technique seems to be very standard in the literature, especially complexity theory but it is interesting to see the application in this context.

Weakness:
- Using MC requires samples to be as uncorrelated as possible to converge. This is often hard to accomplish in practice, hence MCMC.
- There are similarities between this work and some previous work specifically [27].
- Some discussion on the effect of size and degree of circuits seems to be necessary.
- Seems like finding the appropriate circuit that approximates the random variable of interest is not an easy task

---

> ### Author Response · Authors · 2022-08-02
> **Response to Reviewer RDdx**
>
> Thank you for the preliminary review and the questions.
>
> Concerning how to find low-degree circuits, see the responses to other reviewers.
>
> Q1: “how does it compare to others for instance [27], [8]?”
>
> As far as we know, this is the first paper that deals with verifiable Monte Carlo estimation. Our work is based on and goes beyond Williams’s work [27] – our methodological contribution over Williams’s work is in showing how to “compress” the randomness intrinsic in Monte Carlo estimation to obtain resource asymmetry between the verifier and the prover; cf. (2) in our general response to all the reviewers for a more detailed account of our contribution. The interactive proof system framework in Goldwasser et al. [8] by necessitating interaction does not support the outsourcing properties we desire, namely (i) massive distributability of proof preparation through separate evaluations of the proof polynomial, (ii) error-correcting proof preparation by near-linear-time [proof] polynomial interpolation from partially erroneous data, and (iii) deterministic proof preparation enabling third-party verification of the outsourced work; cf. (2).
>
> Q2: “how is pairwise independence not a very restrictive assumption in practice?”
>
> When one aims at __approximation guarantees__, it is common to assume independent samples, which is more restrictive than pairwise independence. This also holds for algorithms based on fast mixing Markov chains (i.e., MCMC): the analysis of the resulting estimator typically assumes independence of the approximate (or less frequently exact) samples from the stationary distribution of the Markov chain. Independence is obtained by simulating the chain from multiple initial states, using independent random numbers, or alternatively by sampling from one chain so that the number of iterations between consecutive samples corresponds to the relaxation time of the chain (or to an upper bound for that). While there are __asymptotic__ converge results for Markov chains, such as the central limit theorem, they do not yield finite-sample guarantees. Only very recently, finite-sample concentration results for Markov chains have been given using martingale techniques (V. Moulos: A Hoeffding inequality for finite state Markov chains and its applications to Markovian bandits. ISIT 2020). MCMC without pairwise independence is currently __not__ a major method for obtaining approximation guarantees.
> If one does __not__ aim at approximation guarantees, it is a common practice to simulate a long Markov chain and form an estimate by taking an average of all samples after a burn-in period. Yes, in this setting the samples are not pairwise independent and our present method is not applicable. But this sort of heuristic MCMC plays no major role when one aims at approximation guarantees, and thus to trustworthiness of not only the delegated computations but also of the underlying estimator.
> In fact, it could be more important to extend our method in another direction: assuming __more than pairwise independence__, not less. Namely, for some Monte Carlo estimators, improved concentration bounds can be proven assuming full independence, or at least k-wise independence for some k larger than 2. We mention this issue briefly in Line 197.
> We will revise Concluding remarks (e.g., Line 372) to further highlight these issues.
>
> Q3: “The performance of an MC estimator depends on the variance of the estimator. How does this method which equips MC estimators with proof correctness depend on the variance of the estimator and how would it be affected?”
>
> We have answered this question in Theorem 5 of the paper. Note that the variance of the estimator is captured by the critical ratio (the second moment divided by the square of the first moment). The theorem says that the effect of the variance is the same whether we consider the baseline estimator or our verifiable extension.
>
> Q4: “Even though the gradient estimation for VI is interesting, however, the used estimate of the gradient is shown to have high variance. How does it work for other more advanced gradient estimators and how would the size of sub-circuits change to keep the error low?”
>
> We considered the basic gradient estimator and the specific Bayesian model because they already present new challenges not covered by the other examples: sampling from a Normal distribution and the presence of non-polynomial functions (exp, ln). While we have not yet taken a deeper look at more advanced gradient estimators, we would expect similar challenges as well as possibilities to address the challenges. For example, estimators based on weak derivatives (measure-valued gradients) would also require (sub)circuits for sampling from Weibull and double-sided Maxwell distributions (see Table 1 of Mohamed et al. [23]). We leave finding efficient approximate circuits for such estimators for future work.

---

### Official Review · Reviewer_LhhL · 2022-07-11

**Rating:** 7
**Confidence:** 2
**Soundness:** 4 excellent
**Presentation:** 3 good
**Contribution:** 4 excellent

**Summary:**

Monte Carlo integration methods are ubiquitous in modern computational tasks and can be highly parallelized across multiple servers. This paper introduces a framework based in algebraic circuits for verifying that the returned estimates provided by the servers were not adversarially manipulated. If the estimates were properly computed, the verification technique will not indicate the information has been manipulated. If they were not properly computed, the technique will indicate that manipulation has occurred with high probability. The authors demonstrate their technique on several classical problems, such as permanent estimation.

**Questions:**

Nothing additional.

**Limitations:**

The authors discuss the limitations of their work in the discussion section at the end of the paper. In particular, they note that it is non-obvious how to construct these verification systems for problems in which estimates do not have pairwise independence, such as MCMC or sequential inference problems. Moreover, they mention that engineering low degree circuits by hand is difficult, and that learning circuit structure is hard even for restricted problem classes.

**Strengths And Weaknesses:**

Strengths: While this paper is mostly outside of my area of knowledge, it seems original and to have practical applications in distributed computing. A strength of the paper is the focus on examples. The examples considered are common, practical problems and illustrate the usefulness of this technique. Another strength is that the introduction clearly illustrates the motivation for the problem and provides insight into the technique being used.

Weaknesses: One weakness is that the main theorem (theorem 1) feels buried in the paper. Most of the paper focuses on example problems, but it may be valuable to give a brief proof sketch of the main theorem.

---

> ### Author Response · Authors · 2022-08-02
> **Response to Reviewer LhhL**
>
> Thank you for the positive review.
>
> We would like to note that we already give a proof of Theorem 1 in the supplement. As the essence of the theorem is due to prior work, our main paper focuses on introducing Theorem 5 and Corollary 6, which contain our key contributions.
>
> Provided that our paper gets accepted, we consider using the allowed one extra page for giving a proof sketch of Theorem 1 in Section 2.

---

### Official Review · Reviewer_xMMk · 2022-07-13

**Rating:** 3
**Confidence:** 3
**Soundness:** 2 fair
**Presentation:** 1 poor
**Contribution:** 2 fair

**Summary:**

  This paper proposes to verify Monte Carlo algorithms through the use of low-degree arithmetic circuits.
  The idea being that if a Monte Carlo algorithm can be made to agree with a computation as represented
  by this circuit we can be reasonably confident in the correctness of the computation after a relatively
  small number of problem instances.


**Questions:**

What inference problems is this method best suited for?

How might one go about finding an arithmetic circuit for a particular problem?

How much additional computation should one expect on a typical example problem?

**Limitations:**

The authors do discuss some of the limitations of this work. Possible negative societal impacts are not discussed, but I suspect the risks are fairly minimal. The research problem being addressed is very general.

**Strengths And Weaknesses:**

  Originality:

  While I'm not if arithmetic circuits have been used to confirm the correctness of Monte Carlo inference,
  the idea of using multiple inference algorithms to check the correctness of each algorithm against
  the other is not new. Many of the examples used in the paper appear to be effectively doing just that. It
  appears the polynomial looks like a tractable expectation.

  Technical Quality:

  While the paper has many examples of how these circuits could be constructed there isn't much that
  looks like a generic method. There are no experiments or really any results showing empirically how
  much extra time is needed to produce a correctness certificate, even for all the examples mentioned.
  The work would greatly benefit from a more thorough exploration of some of the example circuits but
  also a more systematic way to produce these circuits.

  Clarity:

  I found the paper challenging to follow. There are many examples introduced and many extensions suggested
  for the approach, but not much detail about the general algorithm itself.

  Significance:

  I think there is much potential value in producing correctness proofs for Monte Carlo inference algorithms.
  While it's not fully clear how often an arithmetic circuit can be found for all expectations, if that set
  of problems can be better clarified, this could be a big deal.

---

> ### Author Response · Authors · 2022-08-02
> **Response to Reviewer xMMk**
>
> Thank you for the preliminary review and the questions.
>
> The reviewer’s summary of the paper’s ideas and contributions appears to be incorrect. The reviewer writes “the idea of using multiple inference algorithms to check the correctness of each algorithm against the other is not new”. However, the paper’s very idea is to __avoid doing that__: the verifier must run considerably faster than running any single inference algorithm. Our contribution is to show how to achieve that for Monte Carlo estimators. We give __generic methods__ in Theorem 1 and 5, and in Corollary 6 – since the problem is in itself very generic, the methods __necessarily__ take a form of a reduction: we show how to construct an efficiently verifiable estimator, __given__ a low-degree arithmetic circuit for the random variable of interest. These reductions are non-trivial and constitute the key contribution of the paper.
>
> Q1: “What inference problems is this method best suited for?”
>
> This is a good question. Our method is best suited for inference problems that admit Monte Carlo estimators, in which the required number of samples is relatively large compared to the complexity of producing one sample. Ideally, the sample can be computed (exactly or approximately from random input bits) by a low-degree multivariate polynomial, for example, by means of matrix multiplication and determinants. While the baseline estimator is usually derived assuming independent samples, from a strict mathematical standpoint, control on second moments, and thus pairwise independence suffices, which is crucial for our method.
>
> There are numerous inference problems for which our method potentially is suitable. The present paper provides a generic, conceptual and mathematical framework. Naturally, applications to various concrete inference problems are left for future work.
>
> Q2: “How might one go about finding an arithmetic circuit for a particular problem?”
>
> Finding a good arithmetic circuit is akin to finding a good algorithm: we have design paradigms and techniques, but no universal method. For this reason, we equipped the paper with a diverse set of concrete example problems to exhibit the challenges and different ways to address them. That said, we do mention (Lines 127–129) a generic (folklore) way to turn any Boolean circuit to an arithmetic circuit. Since any algorithm (for a fixed-size input) can, in principle, be represented as a Boolean circuit, this is a generic method and one possible starting point – yet, this method is not guaranteed to give a low-degree circuit, and often more insight is needed to arrive at a low-degree circuit design.
>
> Q3: “How much additional computation should one expect on a typical example problem?”
>
> The amount of additional computation is best measured with respect to the total computation time. In Lines 204–205, we write: “Note that the total work T is essentially – up to the factor d and the hidden polylogarithmic factors – the same as the requirement of the baseline algorithm that does not enable verification.” The computational overhead is thus by a factor that is, roughly, proportional to the degree of the arithmetic circuit; in our envisaged use cases (hard problems that require delegation of computations), we would expect the degree to be logarithmic in relation to the total complexity, which is dominated by a relatively large number of samples (cf. our examples of the matrix permanent). It is important to notice that the additional computations can be distributed to multiple machines.
>
> Alternatively, we may consider the computational complexity of the verifier only. If trustworthy computation is required, the baseline is that the verifier does not delegate any computations but does all the needed computations herself. Here our method, of course, does not incur any additional computation – quite the opposite: the verifier obtains considerable computational savings, the computation time being roughly a square root of the baseline. (If trustworthy computation is not pursued, then the “verifier” may delegate nearly all computations and just blindly trust that the received results are correct. Then the additional computations by the verifier, using our method, can be large in relation to the baseline – yet, even then the additional computations of the verifier are small in relation to the total computation time.)

---

> > ### Comment · Reviewer_xMMk · 2022-08-08
> > **Thanks for response**
> >
> > I thank the authors for patiently answering all of my questions. I have a better understanding that the creator of the Monte Carlo estimator also provides the arithmetic circuit which the proof that they correctly implemented the estimator.

---

### Official Review · Reviewer_HG7T · 2022-07-23

**Rating:** 5
**Confidence:** 2
**Soundness:** 2 fair
**Presentation:** 3 good
**Contribution:** 2 fair

**Summary:**

The authors study an interesting problem --- applying an algebraic proof system to certify the correctness of Monte Carlo estimators. They proposed a verifiable randomized approximation scheme based on an arithmetic circuit. Specifically, they consider both the succinct arithmetic circuit and the approximated circuit to approximate random variables. Moreover, they apply their analysis to two applications --- approximate model counting of disjunctive normal form and gradient estimation.

**Questions:**

Could the authors briefly discuss the challenges of deploying their algorithms in practice?

**Ethics Review Area:**

["I don’t know"]

**Limitations:**

None.

**Strengths And Weaknesses:**

The authors propose a generic method to certify the correctness of Monte Carlo estimators with explicit examples. The authors fairly state the potential limitations, e.g., the lack of experiments. Though unfamiliar with this area, I think this is a welcome contribution to the trustworthy machine learning community.

---

> ### Author Response · Authors · 2022-08-02
> **Response to Reviewer HG7T**
>
> Thank you for the preliminary review.
>
> We would like to hear more about the reasoning behind the preliminary scores, particularly for soundness and contribution.
>
> Q1: “Could the authors briefly discuss the challenges of deploying their algorithms in practice?”
>
> The biggest challenge for deploying the algorithms is perhaps the need to handcraft the circuits for each problem separately – given a circuit, our method takes care of the rest. Circuit design is akin to algorithm design – we have design paradigms and various techniques, but no universal design method. Our paper already exhibits various building blocks and tools, but sure there are more to be discovered; we already discuss directions for future research in the Concluding remarks section of the paper. On the other hand, finding an efficient circuit for the random variable of interest need not always be difficult: for instance, in our example (Section 3.3), where the random variable is a square of a determinant, constructing an efficient circuit is relatively straightforward!

---

### Author Response · Authors · 2022-08-02
**General Response**

TO THE AREA CHAIR AND THE REVIEWERS

We thank the four reviewers for their reports. In addition to giving separate individual responses to each reviewer, here we would like to make two general remarks that we also bring to the attention of the area chair:

1)
Each review has a low confidence score; it appears that not all reviewers have been able to assess the paper for its conceptual and mathematical/algorithmic significance. Thus we believe it may be appropriate – as we do in (2) below – to rephrase and further highlight the contribution of the paper from the mathematical/algorithmic standpoint in what we hope are more accessible terms to enable the reviewers to potentially reassess their position and/or to enable additional reviewers with potentially more pertinent expertise to take a position on the paper.

2)
In terms of the contribution of the paper, we would expect __efficiently verifiable__ outsourcing of Monte Carlo estimation with __provable, not heuristic__ guarantees on the approximation quality of the estimate to be a contribution of potentially broad interest in the Machine Learning & AI community – a contribution at the very foundation of the pursuit towards “trustworthy AI”, which is why we chose to submit to NeurIPS.

Key to understanding the contribution is that, in our proof systems for Monte Carlo estimation,
a)  __verifying the proof__ is much more efficient than preparing an estimate, and
b) __preparing the proof__ takes resources comparable to preparing an estimate with state-of-the-art Monte Carlo techniques.

What makes this contribution nontrivial is precisely this asymmetry of resources for the verifier and the prover. In particular, the verifier does __not__ have the resources to repeat the prover’s computation; yet, __given the proof__, the verifier has sufficient resources to verify, with high probability, that the estimate computed by the prover has the desired approximation guarantee. The reader may want to pause for a moment to appreciate that this is a nontrivial mathematical property: a Monte Carlo estimate with sample size N can be verified (from a nontrivial mathematical digest; that is, the proof polynomial) with resources that scale up essentially in the __square root__ of N only.

To enable this resource asymmetry, we have to resolve a challenge not addressed in prior work on proof systems, namely how to “compress” a Monte Carlo sample of size N into a proof digest of size only essentially the square root of N. In essence, we “compress the randomness” __without losing the approximation guarantee__ of the estimator. We observe that second-moment control suffices for the approximation guarantee and thus it suffices to work with only pairwise independent random variables in the sample to obtain a rigorous approximation guarantee. Pairwise independence in the sample can be obtained with low entropy – so low in fact, that even the severely resource-constrained verifier can supply to the prover(s) all the entropy required for the estimate, making proof preparation __deterministic__. Determinism is a serendipitous property since it enables third-party verification of the proof prepared by the prover; in particular, the prover can demonstrate to a party other than the verifier that the contracted computational work [of preparing the estimate as contracted with the verifier based on verifier-supplied entropy] was completed as contracted, and thus the prover can, for example, take appropriate action to secure compensation for the work done. The feasibility of such low-effort-verifiable outsourcing of Monte Carlo estimation is, we expect, a valuable contribution to the pursuit of trustworthy AI.

In summary, we hope the reviewers take into account the remarks above and re-evaluate their position towards the paper as appropriate. On our part we recognise a need to revise the paper to more clearly highlight the conceptual and mathematical/algorithmic contribution over earlier work as above.

---

### Meta-Review · Area_Chair_T1NG · 2022-08-25

**Recommendation:** Accept
**Confidence:** Less certain

**Metareview:**

Reviewers collectively had relatively low confidence when reviewing this paper. This is not surprising, as the paper reads somewhat more like a theory CS paper, rather than a typical ML paper (even a theory one). The main concerns seemed to be that the work is not super concrete, and fairly speculative. These seem like natural concerns for a TCS-style paper, which the authors have tried to address with many explicit examples, and it seems tough to further address this without completely changing the focus of the paper. I cautiously recommend acceptance with the hope that there will be further exploration of these ideas, perhaps with more practical instantiation.

**Award:**

No

---

### Decision · Program_Chairs · 2022-09-14

Accept